# Repositioning Linagliptin for the Mitigation of Cadmium-Induced Testicular Dysfunction in Rats: Targeting HMGB1/TLR4/NLRP3 Axis and Autophagy

**DOI:** 10.3390/ph15070852

**Published:** 2022-07-11

**Authors:** Hany H. Arab, Alzahraa A. Elhemiely, Azza A. K. El-Sheikh, Hana J. Al Khabbaz, El-Shaimaa A. Arafa, Ahmed M. Ashour, Ahmed M. Kabel, Ahmed H. Eid

**Affiliations:** 1Department of Pharmacology and Toxicology, College of Pharmacy, Taif University, P.O. Box 11099, Taif 21944, Saudi Arabia; 2Department of Pharmacology, Egyptian Drug Authority (EDA), Giza 12654, Egypt; zahraa_elhimeily123@yahoo.com (A.A.E.); drahmedhamdy2007@yahoo.com (A.H.E.); 3Basic Health Sciences Department, College of Medicine, Princess Nourah bint Abdulrahman University, P.O. Box 84428, Riyadh 11671, Saudi Arabia; aaelsheikh@pnu.edu.sa; 4Biochemistry Division, College of Pharmacy, Riyadh Elm University, P.O. Box 84891, Riyadh 11681, Saudi Arabia; hanabio@riyadh.edu.sa; 5College of Pharmacy and Health Sciences, Ajman University, Ajman 346, United Arab Emirates; e.arafa@ajman.ac.ae; 6Center of Medical and Bio-Allied Health Sciences Research, Ajman University, Ajman 346, United Arab Emirates; 7Department of Pharmacology and Toxicology, College of Pharmacy, Umm Al Qura University, P.O. Box 13578, Makkah 21955, Saudi Arabia; amashour@uqu.edu.sa; 8Department of Pharmacology, Faculty of Medicine, Tanta University, Tanta 31527, Egypt; ahmed.kabal@med.tanta.edu.eg

**Keywords:** cadmium, HMGB1, NLRP3 inflammasome, apoptosis, autophagy, linagliptin

## Abstract

Cadmium, a ubiquitous environmental toxicant, disrupts testicular function and fertility. The dipeptidyl peptidase-4 inhibitor linagliptin has shown pronounced anti-inflammatory and anti-apoptotic features; however, its effects against cadmium-evoked testicular impairment have not been examined. Herein, the present study investigated targeting inflammation, apoptosis, and autophagy by linagliptin for potential modulation of cadmium-induced testicular dysfunction in rats. After 60 days of cadmium chloride administration (5 mg/kg/day, by gavage), testes, epididymis, and blood were collected for analysis. The present findings revealed that linagliptin improved the histopathological lesions, including spermatogenesis impairment and germ cell loss. Moreover, it improved sperm count/motility and serum testosterone. The favorable effects of linagliptin were mediated by curbing testicular inflammation seen by dampening of HMGB1/TLR4 pathway and associated lowering of nuclear NF-κBp65. In tandem, linagliptin suppressed the activation of NLRP3 inflammasome/caspase 1 axis with consequent lowering of the pro-inflammatory IL-1β and IL-18. Jointly, linagliptin attenuated testicular apoptotic responses seen by Bax downregulation, Bcl-2 upregulation, and suppressed caspase 3 activity. With respect to autophagy, linagliptin enhanced the testicular autophagy flux seen by lowered accumulation of p62 SQSTM1 alongside upregulation of Beclin 1. The observed autophagy stimulation was associated with elevated AMPK (Ser487) phosphorylation and lowered mTOR (Ser2448) phosphorylation, indicating AMPK/mTOR pathway activation. In conclusion, inhibition of testicular HMGB1/TLR4/NLRP3 pro-inflammatory axis and apoptosis alongside stimulation of autophagy were implicated in the favorable actions of linagliptin against cadmium-triggered testicular impairment.

## 1. Introduction

Cadmium is a ubiquitous environmental contaminant with unavoidable exposure given its widespread use in industrial and agricultural activities [1]. The deleterious health effects of cadmium have been characterized in several body organs, including kidney, liver, and lungs. Notably, the male reproductive toxicity of cadmium has been regarded as a serious concern that leads to infertility [2,3]. Animal studies revealed that cadmium is a testicular toxicant that disrupts the blood–testes barrier, culminating in impaired testicular functioning [4]. In this regard, cadmium exposure impairs spermatogenesis, diminishes the quantity/quality of sperm, and prompts several testicular structural and functional defects [2]. Moreover, germ cell death, Sertoli cell dysfunction, impaired testicular steroidogenesis, and lowered testosterone levels have been regarded as hallmarks of cadmium reproductive toxicity in males [1,4]. Despite the well-established toxicity of cadmium on the testicular tissue, the underlying mechanisms have not been completely clarified.

Ample evidence reinforces the crucial role of inflammation in the pathogenesis of cadmium toxicity. High mobility group box protein 1 (HMGB1) is a nucleosomal pro-inflammatory signal unleashed by activated immune cells and necrotic cells [5]. HMGB1 binds the toll-like receptor 4 (TLR4), triggering robust inflammatory events including excessive production of pro-inflammatory cytokines such as tumor necrosis factor-alpha (TNF-α) [5,6]. Of note, the involvement of HMGB1/TLR4 pathway in the pathology of cadmium-induced testicular dysfunction has not been previously elucidated. In the context of inflammatory events, the inflammasomes play a central role in the pathophysiology of testicular damage [7]. Among several inflammasomes, the nucleotide-binding oligomerization domain (NOD)-like receptor family, pyrin domain-containing 3 (NLRP3) inflammasome is the most characterized in terms of testicular disorders [8]. By responding to several cell stressors such as tissue damage signals and reactive oxygen species (ROS), the NLRP3 inflammasome can be assembled and activated, resulting in activation of the pro-inflammatory cytokines interleukin 1 beta (IL-1β) and interleukin 18 (IL-18) [7,8]. In fact, three distinct components of the multiprotein NLRP3 inflammasome have been demonstrated, including the sensor molecule NOD-like receptor NLRP3 and the adaptor signal apoptosis-associated speck-like protein (ASC), together with caspase 1 as the effector protein [7].

Growing lines of evidence advocate the central role of apoptosis in mediating the toxicity of cadmium. Apoptosis is envisioned as a highly regulated cell death program that is principally driven by the intrinsic (mitochondrial) pathway in cadmium-induced testicular injury [3]. Excessive apoptosis has been characterized as negatively impacting the spermatogenesis process and sperm count [9]. In testicular pathologies, the pro-inflammatory events result in prompting the death of germ, Sertoli, and Leydig cells [10,11]. In perspective, the pathology of cadmium-induced testicular insult has demonstrated that the pro-apoptotic Bcl-2-associated x protein (Bax) is overexpressed, while the anti-apoptotic B cell lymphoma-2 protein (Bcl-2) is downregulated, resulting in excessive testicular cell death and impaired spermatogenesis [3].

Distinct from apoptosis, autophagy has been regarded as an evolutionarily conserved stress adaptation mechanism that favors cell survival and precludes cell death [9]. Indeed, autophagy is a lysosomal catabolic program where the misfolded proteins and damaged cellular organelles are segregated into autophagosomes and carried to lysosomes for degradation. Hence, autophagy rids cells of the toxic dysfunctional cell components [12,13]. Alternatively, evidence also exists that excessive autophagy is envisioned as a cell-death pathway, advocating the notion that autophagy may be a double-edged sword under testicular pathological conditions [14]. Coinciding with these studies, overactive autophagy [15,16], as well as disrupted autophagy flux [17,18], have been elucidated in animal models of cadmium-induced testicular injury. Therefore, supplementary studies are required for exploration of the exact role of autophagy in mediating cadmium toxicity in vivo in rats. Of note, the interplay between autophagy and apoptosis has been established in testicular pathologies, where an activated autophagy program has been reported to curtail the apoptotic cell death, promoting cell survival [19]. Moreover, the crosstalk between autophagy and the NLRP3 inflammasome has been characterized, where an augmentation of autophagy has been linked to suppressed NLRP3 inflammasome actions and dampened inflammation [20].

Inhibition of the dipeptidyl peptidase-4 (DPP-4) has been proven to be an effective tool for suppression of inflammatory responses in several pathological models of testicular injury, such as cisplatin [21] and doxorubicin [22] induced testicular damage. In these models, the beneficial actions of DPP-4 inhibition against male reproductive dysfunction have been characterized, including improvement of testosterone levels, sperm deformity, and testicular histopathology. Indeed, the expression of DPP-4 in the testicular tissues [23] and its involvement in the regulation of several pro-inflammatory chemokines and neuropeptides [24] have been previously established. In this regard, evidence exists that DPP-4 is expressed in immune cells, including monocytes and T-lymphocytes, alongside vascular endothelial cells [22,24]. Equally important, DPP-4 inhibition has been reported to enhance the levels of the stromal cell-derived factor-1 alpha (SDF-1α), a substrate for DPP-4, in the testicular tissue. The enhancement of SDF-1α has been associated with favorable testicular outcomes in toxicant-induced testicular injury. This can be linked to the intensified testicular tissue repair and regeneration by the mobilized stem cells to the damaged site [21].

Linagliptin is a selective DPP-4 inhibitor that is originally utilized as anti-diabetic medication for the treatment of type 2 diabetes mellitus (the chemical structure is demonstrated in Figure 1A). DPP-4 is an exopeptidase enzyme that occurs in two forms, namely, membrane-anchored within tissues, and a soluble form in body fluids including plasma [25]. Interestingly, linagliptin has been demonstrated to inhibit both forms of DPP-4 [26]. Since its approval by the FDA in 2011, linagliptin has revealed a marked safety profile with a low prospect for hypoglycemia [27,28]. Likewise, no hypoglycemia has been reported regarding linagliptin administration in normoglycemic animals, as seen in preclinical models of chronic kidney damage [27], inflammatory bowel disease [29], and an Aβ42-induced Alzheimer model [28]. Beyond its glucose modulating actions, linagliptin has demonstrated remarkable anti-inflammatory actions [27,29] and anti-apoptotic effects [30,31] in several pathological models in rodents. However, the potential ameliorative ability of linagliptin against cadmium-induced testicular injury has not been previously considered. Thus, the present work examines the prospect of linagliptin to attenuate the pathological outcomes and impaired spermatogenesis in cadmium-evoked testicular insult in rats. This may add to the clinical merit of linagliptin in type 2 diabetes mellitus patients with co-existing testicular injury. The current study focuses on the molecular events associated with inflammation, apoptosis, and autophagy, especially HMGB1/TLR4/NLRP3 axis. Conspicuously, normoglycemic animals were used in the present study protocol in order to delineate the actions of linagliptin beyond its glucose-modulating competence. This is particularly imperative in the sense that hyperglycemia in animals has been associated with testicular damage in vivo [32]. Similar experimental approaches have been adopted in normoglycemic rodents, such as the animal models of kidney damage and inflammatory bowel disease, and the Aβ42-induced Alzheimer’s model [27,28,29].

## 2. Results

### 2.1. Linagliptin Administration Improves Serum Testosterone and Rectifies Cadmium-Induced Spermatogenesis Impairment in Testes of Rats

As depicted in Figure 1, cadmium intoxication provoked a marked testicular dysfunction in rats. This was demonstrated by a significant (*p* < 0.01) decline of serum testosterone and testicular coefficient by 62% and 36.8%, respectively, in comparison to control animals. With respect to semen analysis, a significant lowering of sperm count (*p* < 0.05), motility (*p* < 0.001), and viability (*p* < 0.05) by 38.7%, 47.1%, and 37.7%, respectively, was detected in the cadmium-intoxicated group. In the same regard, a significant (*p* < 0.001) increase in sperm abnormalities by 162.3% was demonstrated in the cadmium-intoxicated group. The administration of linagliptin to cadmium-intoxicated animals counteracted these aberrations as demonstrated by a significant augmentation of serum testosterone (*p* < 0.05) by 108.3% and testicular coefficient (*p* < 0.01) by 45.2%. Moreover, linagliptin improved the whole spectrum of sperm parameters seen by a significant increase in sperm count (*p* < 0.05) by 47.1%, motility (*p* < 0.01) by 77%, and viability (*p* < 0.05) by 51.3% along with a significant (*p* < 0.001) decline of sperm abnormalities by 44.1%. Of note, the changes in body weight among all experimental groups were non-significant. These findings reveal the competence of linagliptin for rescuing testicular dysfunction and spermatogenesis disruption rendered by cadmium in rats.

### 2.2. Linagliptin Administration Attenuates Cadmium-Induced Histological Lesions in Testes of Rats

Histologic images in control rats (Figure 2A) demonstrated normal testicular architecture and normal spermatogenesis. In this regard, typical morphology of the seminiferous tubules was evident with well-organized germinal cells showing different stages of development alongside normal basement membrane and interstitial tissue. Similar to control rats (Figure 2B), linagliptin-treated control rats demonstrated active seminiferous tubules without histomorphological aberrations. Cadmium intoxication (Figure 2C) provoked atrophy and disruption of the seminiferous tubules. This was associated with disintegration of the germinal epithelium, marked decline in the spermatogenic cell series, and congestion of the interstitial blood vessels. The administration of linagliptin in cadmium-intoxicated animals (Figure 2D) counteracted these histomorphological aberrations, as demonstrated by normal seminiferous tubules and active spermatogenic series. Yet, persistent widening of interstitial spaces was seen in the linagliptin-treated testicular injury group.

### 2.3. Linagliptin Administration Downregulates Testicular DPP-4 and Augments SDF-1α without Changing Testicular Cadmium Content or Serum Glucose in Rats

Relevant to linagliptin’s mechanism of action, the testicular levels of DPP-4 and SDF-1α were quantified. As depicted in Figure 3, cadmium intoxication prompted a significant (*p* < 0.001) increase in testicular DPP-4 content by 279.4%, alongside a significantly (*p* < 0.01) downregulated expression of SDF-1α protein by 53.2%, in comparison to control animals. The administration of linagliptin in cadmium-intoxicated animals reversed these changes, as demonstrated by a significant (*p* < 0.001) decrease in testicular DPP-4 by 41.1% together with significantly (*p* < 0.01) increased levels of testicular SDF-1α by 142.5%.

With reference to cadmium content in testicular homogenates of rats, cadmium intoxication triggered a significant (*p* < 0.001) spike of testicular cadmium metal content by 676% in comparison to control animals (Figure 3C). Despite the tendency to diminish the cadmium levels in the testes of cadmium-intoxicated animals, the administration of linagliptin elicited a non-significant alteration in cadmium metal content. Of note, the changes in serum glucose among all experimental groups were non-significant. These findings reveal the competence of linagliptin to suppress DPP-4 and enhance SDF-1α protein without affecting the uptake of cadmium to the testes of cadmium-intoxicated rats or provoking hypoglycemia.

### 2.4. Linagliptin Administration Curtails Cadmium-Induced Pro-Inflammatory Events and HMGB1/TLR4/NF-κB Pathway Activation in Testes of Rats

As depicted in Figure 4, cadmium intoxication prompted the activation of the pro-inflammatory responses in the testes of rats. This was demonstrated by a significant (*p* < 0.001) increase in testicular TNF-α alongside a significant decline in the anti-inflammatory IL-10 by 166.4% and 57.3%, respectively, in comparison to control animals. This was affirmed by the significantly (*p* < 0.001) upregulated expression of testicular HMGB1 and its receptor TLR4 alongside the nuclear level of the downstream effector NF-κBp65 by 291.6%, 216.7%, and 167.2%, respectively. Administration of linagliptin in cadmium-intoxicated animals counteracted these inflammatory responses, as demonstrated by a significant (*p* < 0.05) decrease in testicular TNF-α by 34.1.% together with a significant increase (*p* < 0.01) in IL-10 by 102.3%. In the same context, the protein expression of testicular HMGB1 (*p* < 0.05), TLR4 (*p* < 0.01), and NF-κBp65 (*p* < 0.01) were significantly downregulated by 36.3%, 35.3%, and 38.9%, respectively. These findings reveal the contribution of inflammatory-signal dampening and HMGB1/TLR4/NF-κB pathway inhibition by linagliptin, at least partly, in the improvement of the testicular damage provoked by cadmium in rats.

### 2.5. Linagliptin Administration Impedes Cadmium-Induced NLRP3/caspase-1/IL-1β Axis Activation in Testes of Rats

As depicted in Figure 5 and Figure 6, cadmium intoxication prompted the activation of NLRP3/caspase-1/IL-1β axis in the testes of rats. This was shown by a significant (*p* < 0.001) increase in testicular NLRP3 protein expression by 474.9% as detected by immunohistochemistry (Figure 5). This was further revealed by a significant (*p* < 0.001) increase in testicular caspase 1 activity alongside IL-1β and IL-18 levels in the testicular tissues by 222.4%, 181.1%, and 235.5%, respectively (Figure 6). Administration of linagliptin in cadmium-intoxicated animals suppressed the testicular NLRP3/caspase-1/IL-1β axis, as demonstrated by a significant decrease in testicular NLRP3 protein expression (*p* < 0.001) (Figure 5) and caspase 1 activity (*p* < 0.01), alongside IL-1β (*p* < 0.01), and IL-18 (*p* < 0.01) levels by 43.2%, 39.3%, 42.4%, and 42.8%, respectively (Figure 6). These data reveal the involvement of NLRP3/caspase-1/IL-1β axis suppression by linagliptin, at least partly, in the amelioration of the testicular damage provoked by cadmium in rats.

### 2.6. Linagliptin Administration Curbs Cadmium-Induced Pro-Apoptotic Events in Testes of Rats

As depicted in Figure 7 and Figure 8, cadmium intoxication prompted activation of the intrinsic apoptosis program in the testes of rats. This was revealed by a significant (*p* < 0.001) increase in the executioner caspase 3 activity by 232% in comparison to control animals. This was affirmed by a significant (*p* < 0.05) increase in Bax immunostaining, a well-characterized pro-apoptotic signal, by 62.1% (Figure 7). Concurrently, the protein expression of the anti-apoptotic signal Bcl-2 was significantly (*p* < 0.05) downregulated by 53%, as illustrated in Figure 8. Administration of linagliptin in cadmium-intoxicated animals reversed these apoptotic events, as demonstrated by a significant decrease in caspase-3 activity (*p* < 0.01) and Bax immunostaining (*p* < 0.05) by 37.2% and 33.3%, respectively (Figure 7). In the same direction, the protein expression of Bcl-2 was significantly (*p* < 0.01) upregulated by 162.5% (Figure 8). These data reveal the involvement of apoptosis suppression by linagliptin, at least partly, in the attenuation of the testicular damage provoked by cadmium in rats.

### 2.7. Linagliptin Administration Enhances Cadmium-Induced Autophagy in Testes of Rats

As depicted in Figure 9, cadmium intoxication triggered a compromised autophagy response in the testes of rats. This was demonstrated by a significant (*p* < 0.001) increase in testicular protein expression of p62 SQSTM1, a protein marker that is degraded by autophagy, by 278.1%, in comparison to control animals, suggesting its accumulation due to an impaired autophagy flux. Concurrently, the protein expression of Beclin 1, an autophagy signal that is required for initiation of autophagosome formation, displayed a significant (*p* < 0.01) decline by 56.8% in cadmium-intoxicated rats. Administration of linagliptin in cadmium-intoxicated animals afforded reversal of these events and enhancement of the autophagy response in the testes of cadmium-intoxicated rats. This was verified by a significant (*p* < 0.01) lowering in p62 SQSTM1 protein expression by 37.3% together with a significant (*p* < 0.01) upregulation of Beclin 1 protein expression by 106.1%. These data disclose the involvement of autophagy stimulation by linagliptin, at least partly, in the mitigation of the testicular damage provoked by cadmium in rats.

### 2.8. Linagliptin Administration Counteracts Cadmium-Induced AMPK/mTOR Pathway Inhibition in Testes of Rats

As depicted in Figure 10, cadmium intoxication triggered suppression of AMPK/mTOR, a pro-autophagic pathway [12,14], in the testes of rats. This was revealed by a significant (*p* < 0.001) lowering in AMPK (Ser487) phosphorylation to the total AMPK protein by 62.9%, in comparison to control animals. Inhibition of the testicular AMPK/mTOR cascade was affirmed by the significant (*p* < 0.001) increase in mTOR(Ser2448) phosphorylation to the total mTOR protein by 228.4%. Administration of linagliptin in cadmium-intoxicated animals counteracted these detrimental events and stimulated the AMPK/mTOR pathway in the testes of cadmium-intoxicated rats. This was demonstrated by a significant (*p* < 0.01) increase in AMPK(Ser487) phosphorylation by 142.7%, alongside a significant (*p* < 0.01) lowering in mTOR(Ser2448) phosphorylation by 42.3%. These data indicate the involvement of AMPK/mTOR pathway stimulation by linagliptin, at least partly, in the attenuation of the testicular damage provoked by cadmium in rats.

## 3. Discussion

The findings of the present study revealed that linagliptin ameliorated cadmium-induced testicular impairment and histopathological changes in vivo in rats by suppression of testicular inflammatory responses and apoptosis alongside stimulation of autophagy. At least partly, curbing of HMGB1/TLR4/NLRP3 axis together with activation of AMPK/mTOR pathway were implicated in the observed favorable actions of linagliptin in the testes of cadmium-intoxicated rats (Figure 11). Notably, these favorable outcomes are mainly ascribed to DPP-4 inhibition, which elicits favorable anti-inflammatory and immune-modulatory actions [21,22]. So far, linagliptin has been reported as the most potent inhibitor of the DPP-4 enzyme among the family of gliptins, and it has a large volume of distribution, resulting in effective tissue penetration and tight-binding of the membrane-anchored DPP-4 [26].

The involvement of the pro-inflammatory HMGB1/TLR4 pathway in the pathology of testicular dysfunction has been previously established [5]. Yet, its role in cadmium-triggered testicular injury has not been explored. In testicular pathology, the damage-associated molecular pattern HMGB1 is actively secreted by immune cells or passively released by necrotic cells to amplify the inflammatory reactions [5,34]. HMGB1 can signal through several receptors, particularly the TLR4 receptors under intensified pro-inflammatory immunological responses, culminating in excessive cytokine production and chemotaxis [34]. In testicular macrophages, the interaction between HMGB1 and TLR4 has been demonstrated, an event that triggers NF-κB pathway stimulation and associated upregulated expression of proinflammatory cytokines [5,34]. In testes of infertile men and an autoimmune orchitis rat model, activation of HMGB1 signaling has been characterized [5]. In line with these data, the present work revealed evident testicular pro-inflammatory responses and activation of the HMGB1/TLR4 cascade in response to cadmium exposure. These pro-inflammatory responses were attenuated by linagliptin administration. Likewise, targeting HMGB1/TLR4 pathway and associated pro-inflammatory cytokines by ethyl pyruvate has been regarded as an encouraging approach to mitigate testicular disruption and impaired spermatogenesis in experimental autoimmune orchitis [5]. Consistently, the downregulated expression of HMGB1 has been reported to activate autophagy flux through Beclin 1-dependent pathway [35,36], an event that also dampens apoptosis [5,37]. Of note, these findings coincide with the reported HMGB1 downregulation by linagliptin in chondrogenic ATDC5 cells [38] and an animal model of inflammatory bowel disease [29]. Moreover, the marked anti-inflammatory actions of linagliptin have been characterized in an endotoxin-induced acute kidney injury model [39], where DPP-4 inhibition inhibited NF-κB pathway and the downstream pro-inflammatory cytokines IL-1β and TNF-α.

The NLRP3 inflammasome is composed of three defined components, including the NLRP3 as the ROS sensor, the ASC subunit as the adapter component, and the effector caspase 1. A two-step process has been identified for the regulation of NLRP3 inflammasome. First, danger-associated molecular patterns (DAMP) or ROS provide the “priming” stimulus, which results in the upregulated expression of the three components before their assembly [7,12]. A well-characterized example of the priming step is the binding of DAMP to TLR4 receptors [7,20]. Then, the activating stimulus provokes assembly of the active NLRP3 complex, culminating in the production of pro-inflammatory IL-1β and IL-18 from their inactive precursors pro-IL-1β and pro-IL-18, respectively [12]. In line with these concepts, the current study showed the testicular activation of NLRP3 inflammasome seen by the upregulated expression of NLRP3, enhanced caspase 1 activity, and the increased production of IL-1β and IL-18. These pro-inflammatory events were attenuated by linagliptin. In testicular pathologies, the literature established the crosstalk between NLRP3 and the autophagy process. In this regard, the modalities that switch on testicular autophagy flux have been demonstrated to dampen the activity of the NLRP3 inflammasome, culminating in attenuated testicular damage [20]. Notably, these findings are in line with the reported NLRP3 inflammasome inhibition by linagliptin in a rodent model of myocardial infarction [40]. In this model, linagliptin suppressed the NLRP3/ASC pathway, TLR4 expression, and the levels of the downstream pro-inflammatory cytokines, including IL-1β and TNF-α. Likewise, similar mechanisms have mediated the anti-inflammatory actions of linagliptin in a doxorubicin-induced nephropathy model in rats, resulting in enhanced renal functional outcomes [41].

Apoptosis of germ cells in testes of rodents has been linked to impaired spermatogenesis and associated disruption of sperm count and viability [4,10]. Indeed, excessive proinflammatory events culminate in the initiation of apoptosis machinery in testes of cadmium-intoxicated animals [4,10,11,14]. The crucial role of the intrinsic apoptosis pathway in the pathogenesis of cadmium-induced testicular dysfunction has been affirmed. In this regard, apoptosis of germ cells and Leydig cells has been linked to disrupted spermatogenesis [11,42]. In line with these findings, the present study revealed that cadmium exposure instigated testicular apoptosis with upregulation of the pro-apoptotic Bax and downregulation of the anti-apoptotic Bcl2 protein expression, culminating in enhanced activity of executioner caspase 3, as previously reported in cadmium intoxication [14]. Favorably, these pro-apoptotic events were dampened by linagliptin in favor of cell survival. Consistently, the observed stimulation of autophagy by linagliptin likely advocates the survival of testicular cells as demonstrated by the antagonistic role of autophagy against apoptosis in germ cells, Leydig cells [15], and spermatocytes [43]. In the same context, evident anti-apoptotic effects of linagliptin have been described in a diabetic nephropathy model in db/db mice [30] through dampening caspase 3 activity and upregulation of the anti-apoptotic Bcl2. Given the notion that apoptosis is initiated by excessive pro-inflammatory signals [10,11,14], the present data together with the reported anti-inflammatory features of linagliptin [27,29] may, at least partly, mediate its anti-apoptotic actions.

A major finding in the present work is that cadmium exposure impaired testicular autophagy in testes of rats. This was supported by the accumulation of p62 SQSTM1, a negative autophagy marker that denotes the blockade of autophagosome degradation and impaired autophagy flux [14,44]. This is perceived from the fact that p62 SQSTM1 is selectively incorporated into autophagosomes, and p62 SQSTM1 itself is degraded following the fusion of autophagosomes with lysosomes [9,14,44]. Concurrently, cadmium exposure triggered Beclin 1 upregulation, a critical regulator of autophagosome sequestration [14] in testes of rats. In cadmium intoxication models, impaired autophagy flux [17,18], as well as overactive autophagy [15,16] have been characterized, pointing to the need for supplemental studies for determining the exact role of autophagy in the pathology of testicular dysfunction provoked by cadmium. In line with the present findings of impaired testicular autophagy, the literature has revealed that defective autophagy impacts the later stages of spermatogenesis, resulting in defective acrosome biogenesis, spermatogenesis suppression, and rodent infertility [17,18]. In the same regard, excessive germ cell apoptosis, spermatogenic failure, and accumulation of damaged mitochondria have been characterized by defective testicular autophagy in rodents [9]. Conceptually, accumulation of damaged mitochondria, the principal ROS generator, has been linked to initiating apoptotic cell death and associated activation of caspase 3 [12]. The later event degrades the autophagy signal Beclin 1, resulting in the blockade of autophagy flux [45].

Accumulating evidence revealed that therapeutic agents that target germ cell autophagy stimulation have been associated with favorable outcomes in animal models of cadmium-evoked testicular dysfunction [17,18,19]. Indeed, the functional role of autophagy in advocating cell survival in response to cell stressors has been reinforced [44,46,47]. In this context, autophagy stimulation clears damaged mitochondria and compromised macromolecules, thereby promoting cell survival and rescuing cells against cell death [12,15]. In line with these facts, the current work demonstrated autophagy stimulation in response to linagliptin evidenced by decreased p62 SQSTM1 accumulation together with increased Beclin 1 levels. These observations coincide with linagliptin’s pro-autophagic effects that have been reported in a diabetic nephropathy model in db/db mice by upregulating Beclin 1 expression, culminating in marked renoprotection [30]. In the context of steroidogenesis, testicular autophagy stimulation has been evidenced as an effective mechanism for augmented steroidogenesis and associated boosting of testosterone levels [12]. This is interceded by enhanced cholesterol trafficking to Leydig cells via the autophagosome system [48,49]. Likewise, autophagy augmentation is associated with enhanced acrosome biogenesis in sperm [20].

Relevant to testicular autophagy stimulation, the present work revealed that linagliptin activated AMPK/mTOR pathway, an event that drives autophagy flux boosting in testicular pathologies [12,14,20]. Indeed, the involvement of AMPK/mTOR stimulation has been described to attenuate cadmium-evoked testicular dysfunction [18]. In fact, AMPK activation through its phosphorylation is a positive signal for autophagy flux, an event that boosts cellular energy generation from recycled proteins [12,50]. Moreover, lowered phosphorylation of the negative autophagy signal mTOR has been linked to autophagy activation [50]. Conspicuously, activation of AMPK has been previously characterized by linagliptin in experimental models of endotoxin-induced acute kidney injury [39], myocardial infarction [51], and neointima hyperplasia in diabetic rats [52].

## 4. Materials and Methods

### 4.1. Chemicals

Linagliptin (Trajenta^®^) was supplied by Boehringer Ingelheim (Ridgefield, CT, USA), while the testicular toxicant cadmium chloride (Cat. # 202908) and other chemicals for biochemical determinations were purchased from Sigma-Aldrich (Sigma-Aldrich Chemical Co., St. Louis, MO, USA).

### 4.2. Animals

Twenty-four male 12-week-old Wistar albino rats weighing 200–220 g were procured from the animal facility of the Egyptian Drug Authority (EDA). Housing of animals was established in plastic cages with wood-chip animal bedding under 12 h light/12 h dark cycles and 22–24 °C ambient temperature. The rats were granted free access to drinking water and rodent food pellets alongside a 10-day acclimatization span.

The handling of animals and all relevant procedures for animal treatment were conducted in agreement with the guidelines of the National Institutes of Health (NIH) Guide for the Care and Use of Laboratory Animals (USA, Publication No. 85-23, revised 1996). Approval of the current experimental protocol was endorsed by the EDA’s Research Ethics Committee under the ethical approval code NODCAR/13.

### 4.3. Experimental Design and Protocol

Animals were distributed to experimental groups randomly by a technician unaware of the design of the study. As depicted in Figure 12, the first group (Control group; 6 animals) received a daily gavage of the vehicle of cadmium chloride (normal saline; 10 mL/kg) plus a daily gavage of linagliptin vehicle (0.5% carboxymethyl cellulose; CMC; 10 mL/kg, 1 h after cadmium chloride vehicle) for 8 weeks. The second group (Control + LIN group; 6 animals) received a daily gavage of normal saline vehicle (10 mL/kg) plus a daily gavage of linagliptin (5 mg/kg; 1 h after cadmium chloride vehicle) suspended in CMC (10 mL/kg) for 8 weeks. The third group (Cd group; 6 animals) received a daily gavage of cadmium chloride (5 mg/kg in normal saline; 10 mL/kg) plus a daily gavage of CMC (1 h after cadmium chloride solution) for 8 weeks. The fourth group (Cd + LIN group; 6 animals) received a daily gavage of cadmium chloride (5 mg/kg in normal saline; 10 mL/kg) plus a daily gavage of linagliptin (5 mg/kg; 1 h after cadmium chloride solution suspended in CMC (10 mL/kg) for 8 weeks. The current experimental protocol is in agreement with the published literature [18,53,54]. The chosen dose of linagliptin was selected according to previous reports that elucidated its competence for the mitigation of several experimental pathologies [28,55,56,57].

On day 60 and after overnight fasting, animals were weighed, and the blood samples were collected by a retro-orbital puncture to the venous plexus under 30 mg/kg thiopental sodium anesthesia (i.p.). Serum separation was carried out by centrifugation at 1000× *g* for 10 min, and testosterone levels were determined. Animal euthanization was applied under thiopental anesthesia by cervical dislocation. Immediately, the testes and cauda epididymis were dissected out and washed with normal saline, and the testicular coefficient was obtained from the following formula: the 2 testes weight (g)/total body weight of rat (kg). One testis was preserved in a 10% neutral buffered formalin solution (3 animals chosen randomly from each group). The other testis was stored at −20 °C for the biochemical assays. To this end, part of the testis was homogenized in RIPA buffer provided with a cocktail of protease/phosphatase inhibitors for ELISA determinations. Then, centrifugation at 10,000× *g* for 20 min at 4 °C was carried out, and the supernatant was used. The cauda epididymis was utilized for sperm parameters. 

### 4.4. Semen Analysis

Analysis of sperm count, motility, viability, and abnormality was applied as reported [18]. To this end, mincing of the cauda epididymis in normal saline was applied, and sperm counting was implemented using a hemocytometer at 400× magnification under a light microscope. The sperm motility was executed at 37 °C within 4 min of semen extraction. This was done by mixing one drop of semen suspension with one drop of sodium citrate (2.9%) onto a preheated glass slide, and the percentage of motile sperm was determined under a light microscope at 200× magnification. The percentage of sperm abnormal morphology was determined by detecting the sperms with abnormal head and tail. Finally, the sperm viability was examined after staining the sperm suspension with eosin–negrosin stain, and recording the unstained viable sperm was conducted using a hemocytometer at 400× magnification under a light microscope.

### 4.5. Measurement of Serum Testosterone and Glucose and Testicular DPP-4 and SDF-1α

Serum testosterone levels were quantified by a Cusabio ELISA kit (Cusabio Technology, Houston, TX, USA; Catalog # CSB-E05100r) as directed by the provider. The final color was measured at 450 nm. The serum glucose levels were measured by a colorimetric assay kit (HUMAN Diagnostics, Germany, Catalog # BD184). The final color was read at 500 nm. The content of DPP-4 in the testicular tissue homogenate was measured by a rat-specific ELISA kit (MyBioSource, San Diego, CA, USA, Catalog # MBS700649). The SDF-1α content in the testicular homogenate was quantified by an Elabscience ELISA kit (Elabscience, Wuhan, China; Catalog # E-EL-R3027) as directed by the provider. The final color was read at 450 nm.

### 4.6. Determination of Testicular Cadmium Levels

The assay of cadmium metal content in the testicular homogenate was implemented as previously demonstrated [58]. First, digestion of the testicular tissue in 1 M nitric acid was applied. Then, ashing of the resulting digestion solution at 150 °C was implemented. With the aid of a PerkinElmer 3100 atomic absorption spectrophotometer, the cadmium metal was detected at 228.8 nm.

### 4.7. Histological Analysis and Scoring of Histological Damage

A routine histological protocol was adopted for examining the microscopic changes of the testicular tissue and the spermatogenesis process in rat testis as depicted in [59,60]. The fixed testes (in 10% neutral buffered formalin) were embedded in paraffin blocks and cross-sectioned at a right angle to the longest axis of the testes at 5 µm thickness. Following the deparaffinization process and rehydration, staining of sections was applied using hematoxylin and eosin (H&E) stain. Slides were examined by an experienced technician under a light microscope (Leica Microsystems GmbH, Wetzlar, Germany). The photomicrographs were captured with the aid of a digital camera fitted to the light microscope. Slide analysis and photomicrograph capturing were implemented in a blinded manner to exclude bias.

### 4.8. Immunohistochemical Staining of Testes

Paraffin-embedded tissue sections were immunostained to detect the testicular protein expression of NLRP3, Bax, and Bcl-2, as previously reported [61]. To this end, dewaxing, rehydration, and antigen retrieval of tissue sections by boiling in citrate buffer (pH 6.0) were implemented. Quenching of endogenous peroxidase activity was performed with a 3% hydrogen peroxide solution for 20 min. Blocking of sections with 5% bovine serum albumin was done in a humidified chamber. Thereafter, the sections were incubated overnight at 4 °C with specific primary antibodies against NLRP3 (1:100 dilution; Catalog # GTX00763, GeneTex Inc., Irvine, CA, USA), anti-Bcl-2 (1:100 dilution; Catalog # PA1-30411), or anti-Bax (1:100 dilution; Catalog # 33-6600; Thermo Fisher Scientific, Waltham, MA, USA). After washing, the sections were incubated with HRP-tagged secondary antibody for 30 min. The target proteins were visualized with the aid of diaminobenzidine (DAB) with subsequent counterstaining of sections with hematoxylin. Immunohistochemical micrographs were inspected under a light microscope provided with a digital camera (Leica Microsystems GmbH, Wetzlar, Germany). Three specimens were randomly selected from each experimental group, and 6 non-overlapping fields were examined at 400× magnification and photographed for each specimen. Quantification of the brown immunostaining of target proteins was carried out using the Leica Application Module. This was expressed by the area of the brown stain to the full area of the field. The specificity of the used primary antibodies was checked by preparing negative control sections with normal rabbit serum instead of primary antibodies.

### 4.9. Determination of Testicular Caspase-1/IL-1β and HMGB1/TLR4/NF-κB Pathways and Inflammatory Mediators

The activity of caspase-1 was quantified in the testicular homogenate using a colorimetric kit (R&D Systems, MN, USA, Catalog # K111-100). The final color was measured at 405 nm. As directed by the manufacturer, the results were presented as the fold-change of the data means. The quantification of the pro-inflammatory cytokines in the testicular homogenate was implemented with the aid of specific ELISA kits for IL-1β (Cusabio Technology, Houston, TX, USA; Catalog # CSB-E08055r), IL-18 (Cusabio Technology, Houston, TX, USA; Catalog # CSB-E04610r), TNF-α (Catalog # RTA00), and IL-10 (Catalog # R1000; R &D systems incorporation, MN, USA). The intensity of the final color was read at 450 nm. Commercial ELISA kits were used for the measurement of TLR4 (Cusabio Technology, Houston, TX, USA; Catalog # CSB-E15822r), HMGB1 (Elabscience, Wuhan, China; Catalog # E-EL-R0505), and NF-κBp65 (Elabscience ELISA kit, Wuhan, China; Catalog # E-EL-R0674) in the testicular homogenate. The O.D. of the final color was measured at 450 nm. The NF-κBp65 was measured in the nuclear extract of the testicular homogenate. This was prepared with the aid of a Cayman nuclear extraction kit (Cayman Chemical Company, Ann Arbor, MA, USA; Catalog # 10009277).

### 4.10. Measurement of Testicular Autophagy Markers

The Beclin 1 and SQSTM-1/p62 autophagy markers were quantified in the testicular homogenate with specific ELISA kits (Cusabio Technology, Houston, TX, USA, Catalog # EL002658RA and MyBioSource, San Diego, CA, USA, Catalog # MBS3809397, respectively), as directed by the manufacturer. The axis of AMPK/mTOR was determined in the testicular homogenate by detecting the phosphorylation of AMPK and mTOR, respectively [62]. To this end, quantification of p-AMPK(Ser487) and total (pan) AMPK protein level was determined with the aid of a specific ELISA kit (RayBiotech., Peachtree Corners, GA, USA; Catalog # PEL-AMPKA-S487-T). Half of the wells of the ELISA plate were pre-coated with a specific primary antibody against the phosphorylated form of AMPK, whereas the other half of the wells was precoated with a specific primary antibody against the total form of AMPK. Likewise, quantification of p-mTOR(Ser2448) and total (pan) mTOR protein levels was determined with the aid of specific ELISA kits (Cell Signaling Technology, Danvers, MA, USA, Catalog # 7976C and Catalog #7974C, respectively). For these kits, measurement of the O.D. of the final color was done at 450 nm.

### 4.11. Measurement of Testicular Pro-Apoptotic Caspase-3 Activity

A commercial colorimetric kit (Sigma-Aldrich, St. Louis, MO, USA, Catalog # CASP-3-C) was employed for the quantification of testicular caspase-3 activity in the testicular homogenate. The O.D. of the final color was read at 405 nm. As disclosed by the manufacturer, the results were presented as the fold-change of the data mean.

### 4.12. Statistics

GraphPad Prism 6 software was used for processing the statistical analysis (GraphPad Software, Inc., San Diego, CA, USA). The data normality was verified by the Shapiro–Wilk test. The data were analyzed with a one-way analysis of variance with Tukey’s post hoc test for the multi-comparisons among experimental groups. The probability of error (*p*-value of <0.05) was kept as the minimum acceptable threshold of statistical significance. The data values were demonstrated as the mean ± standard error of the mean (S.E.M.).

## 5. Conclusions

In summary, the present work yields in vivo evidence of the involvement of the anti-inflammatory, anti-apoptotic, and pro-autophagic features of linagliptin in improving testicular dysfunction provoked by cadmium. These beneficial effects are mainly driven by DPP-4 inhibition, which has been linked to marked anti-inflammatory and immune modulatory effects seen in several experimental testicular injury models [21,22]. In response to DPP-4 inhibition, previous reports revealed that suppression of the pro-inflammatory events by linagliptin is conducted likely through disrupting the interaction between membrane-anchored DPP-4 and integrin β1 alongside the modulation of mi-RNA-29 [26]. In the same regard, DPP-4 inhibition in T-lymphocytes has been demonstrated to disrupt the protein–protein interaction between the cysteine-rich domain of DPP-4 and adenosine deaminase (ADA), culminating in dampened pro-inflammatory cytokine production [63]. In animal models of testicular injury, DPP-4 inhibition has been associated with increased testicular SDF-1α [21] and glucagon-like peptide-1 (GLP-1) [24], resulting in improved testicular functional outcomes. Due to its impact on many biological substrates, the exact molecular mechanisms of DPP-4 inhibition exerted by linagliptin cannot be defined. Hence, additional studies are warranted to delineate the detailed molecular mechanisms of linagliptin. Moreover, further investigations are needed to characterize whether the anti-inflammatory/anti-apoptotic actions of linagliptin are driven by direct inhibition on each component of HMGB1/TLR4/NLRP3 axis or if other intermediate steps are also involved.

## Figures and Tables

**Figure 1 pharmaceuticals-15-00852-f001:**
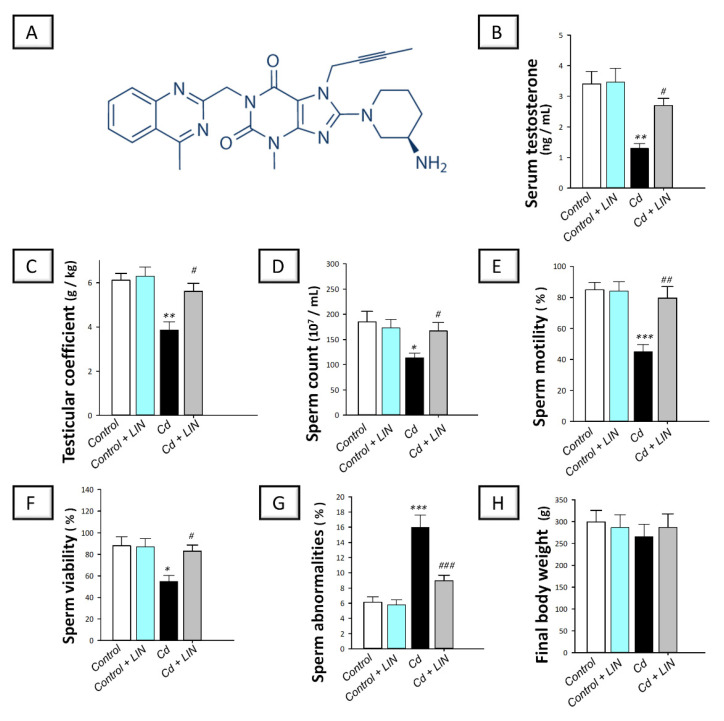
Effect of cadmium and/or linagliptin administration on testicular dysfunction and sperm parameters in rats. The co-treatment with linagliptin (the chemical structure displayed in (**A**) rescued the testicular dysfunction and improved the sperm abnormalities that were instigated by cadmium metal. This was demonstrated by increase of serum testosterone (**B**), testicular coefficient (**C**), sperm count (**D**), sperm motility (**E**), and sperm viability (**F**), alongside lowering sperm abnormalities (**G**), without affecting the final body weight (**H**). ** p* < 0.05, *** p* < 0.01, and **** p* < 0.001 show statistical significance compared to the group of control rats; *^#^ p* < 0.05, *^##^ p* < 0.01, and *^###^ p* < 0.001 show statistical significance compared to the group of cadmium-intoxicated rats. Values are stated as the mean ± standard error of the mean for 6 animals per group. LIN, linagliptin; Cd, cadmium chloride.

**Figure 2 pharmaceuticals-15-00852-f002:**
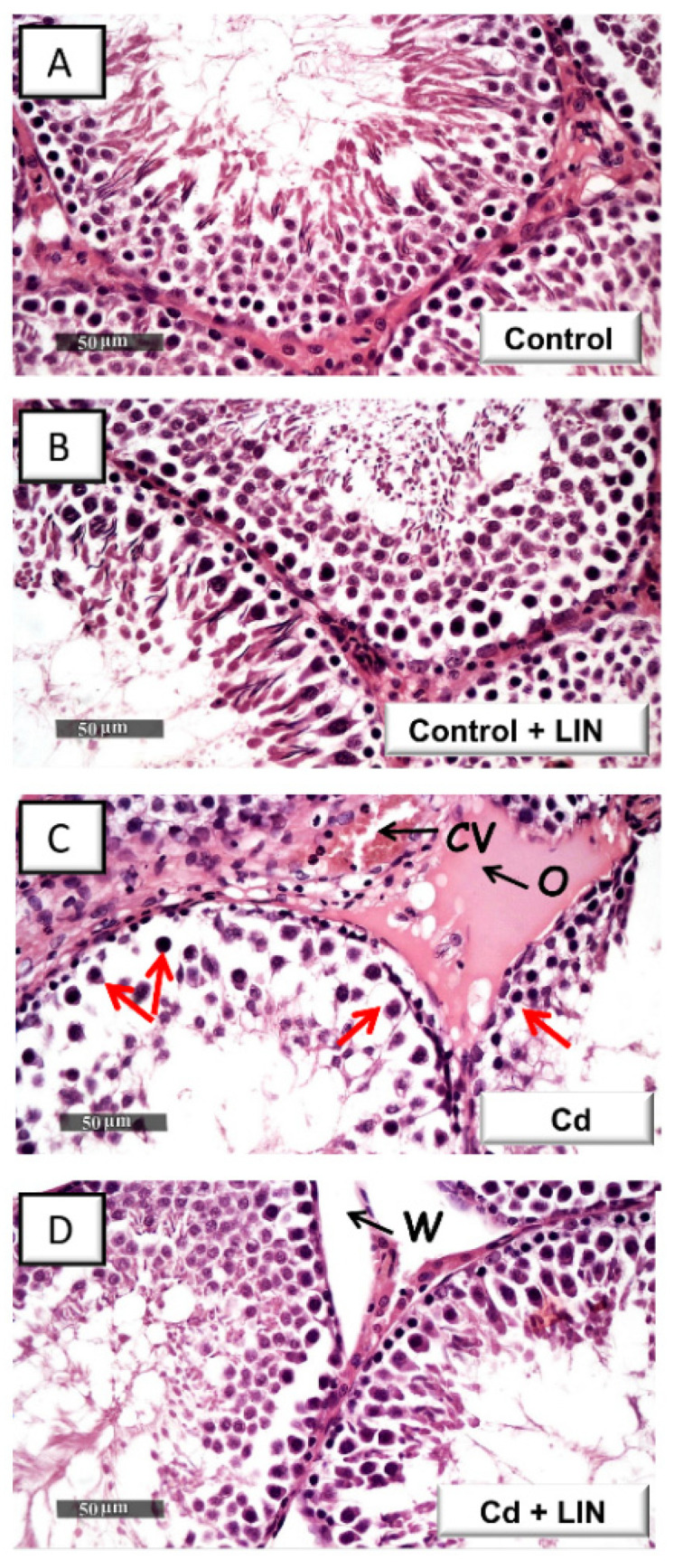
Effect of cadmium and/or linagliptin administration on the testicular histomorphological changes in rats. The testicular cross-sections were stained with hematoxylin and eosin (H&E) stain and examined using light microscopy. Normal morphology of seminiferous tubules and complete spermatogenic series were evident in the control (**A**) and linagliptin-treated control group (**B**). Atrophy of the seminiferous tubules, germinal epithelium degenerative changes (red arrow), moderate interstitial edema (O), and severely congested blood vessels (CV) were observed in cadmium-intoxicated sections (**C**). These histopathological lesions were attenuated in the testicular sections from the linagliptin-treated testicular damage group (**D**). Yet, widening (W) of interstitial spaces was still seen. Three animals were randomly chosen from each experimental group for histopathology. LIN, linagliptin; Cd, cadmium chloride.

**Figure 3 pharmaceuticals-15-00852-f003:**
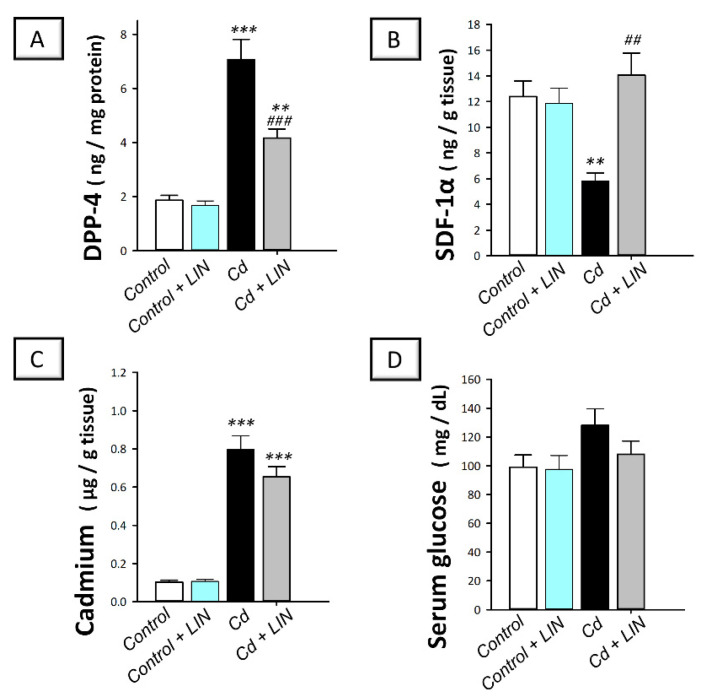
Effect of cadmium and/or linagliptin administration on testicular DPP-4, SDF-1α, and cadmium metal content alongside serum glucose in rats. The co-treatment with linagliptin downregulated testicular dipeptidyl peptidase 4 content (DPP-4; (**A**)) and increased testicular stromal cell-derived factor-1 alpha (SDF-1α; (**B**)) without modifying testicular cadmium metal content (**C**) or serum glucose (**D**) in cadmium-intoxicated rats. *** p* < 0.01, and **** p* < 0.001 show statistical significance compared to the group of control rats; *^##^ p* < 0.01, and *^###^ p* < 0.001 show statistical significance compared to the group of cadmium-intoxicated rats. Values are stated as the mean ± standard error of the mean for 6 animals per group. LIN, linagliptin; Cd, cadmium chloride.

**Figure 4 pharmaceuticals-15-00852-f004:**
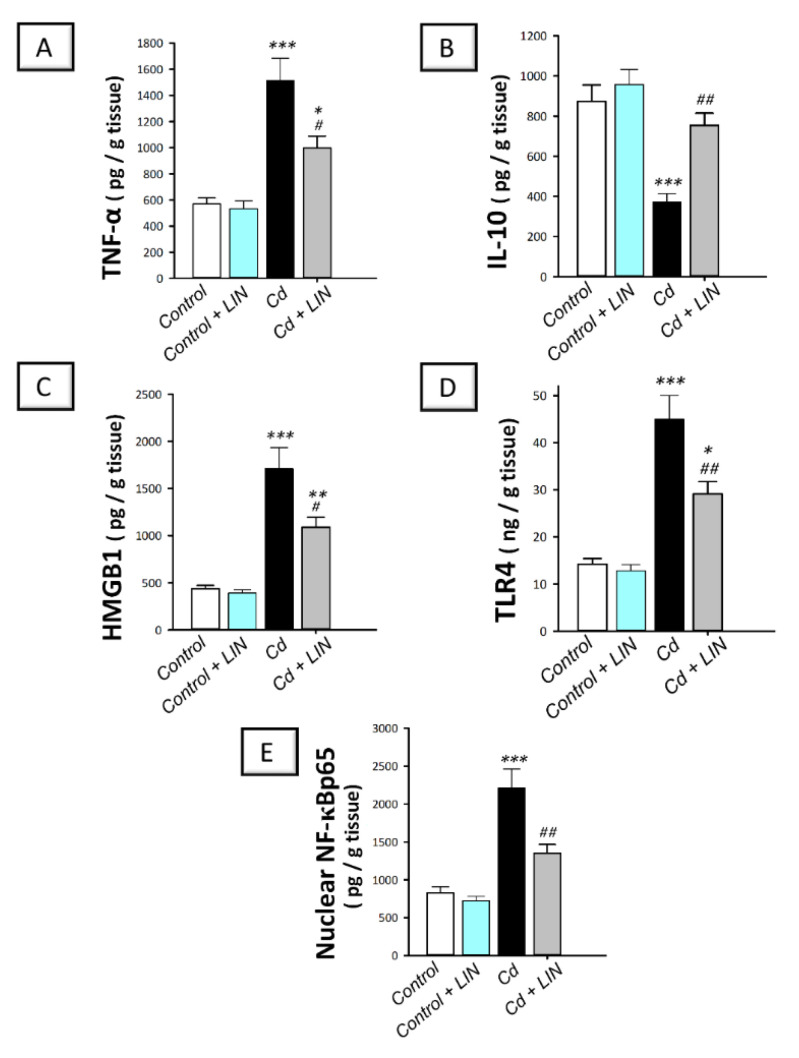
Effect of cadmium and/or linagliptin administration on the testicular pro-inflammatory response and HMGB1/TLR4/NF-κB pathway in rats. The co-treatment with linagliptin lowered testicular TNF-α (**A**) and augmented IL-10 (**B**) alongside curtailing the testicular HMGB1/TLR4/NF-κB pathway that was stimulated by cadmium metal. The later event was demonstrated by downregulated protein expression of the high mobility group box protein 1 (HMGB1; (**C**)), toll-like receptor 4 (TLR4; (**D**)), and nuclear expression of the nuclear factor kappa B (NF-κBp65; (**E**)). ** p* < 0.05, *** p* < 0.01, and **** p* < 0.001 show statistical significance compared to the group of control rats; *^#^ p* < 0.05, and *^##^ p* < 0.01 show statistical significance compared to the group of cadmium-intoxicated rats. Values are stated as the mean ± standard error of the mean for 6 animals per group. LIN, linagliptin; Cd, cadmium chloride.

**Figure 5 pharmaceuticals-15-00852-f005:**
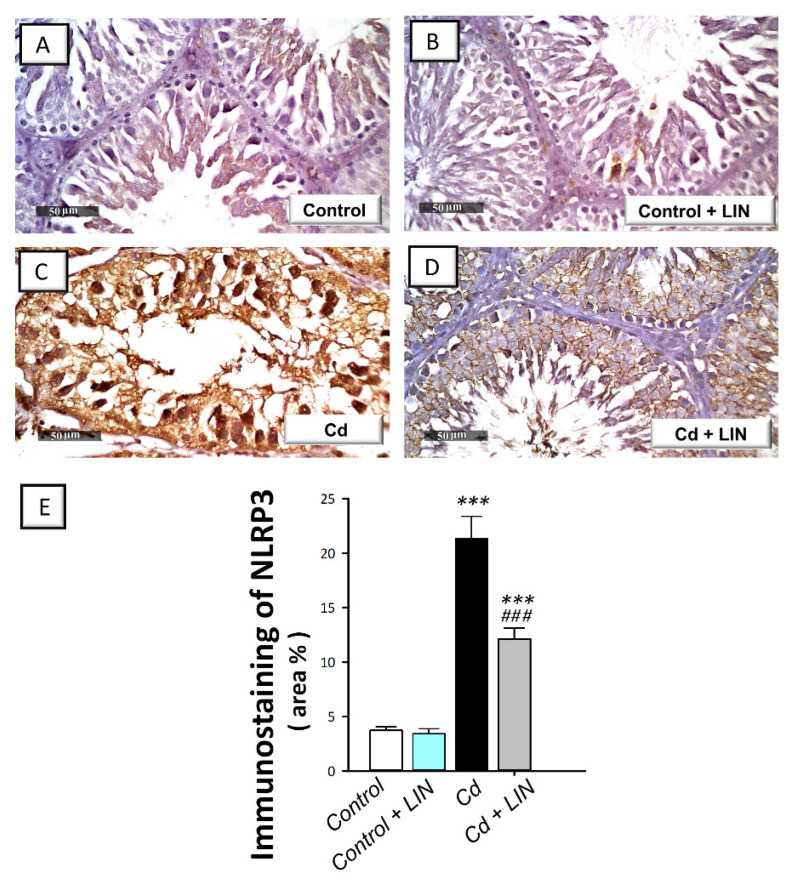
Effect of cadmium and/or linagliptin administration on testicular NLRP3 protein expression in rats. The co-treatment with linagliptin downregulated the protein expression of the nucleotide-binding oligomerization domain (NOD)-like receptor family, pyrin domain-containing 3 (NLRP3) as detected by immunohistochemistry that shows up as brown color at 400× magnification. (**A**,**B**) The immunostaining of testicular NLRP3 protein demonstrates minimal expression in the control and control + LIN groups. (**C**) The immunostaining of the pro-inflammatory NLRP3 was upregulated in the cadmium-intoxicated group. (**D**) The immunostaining of NLRP3 was downregulated in the Cd + LIN group. (**E**) Quantitative analysis of NLRP3 immunostaining in the testicular tissue presented as the area % of NLRP3 immunoreactivity/full microscopic field area. Six non-overlapping fields were inspected for each specimen. **** p* < 0.001 show statistical significance compared to the group of control rats; *^###^ p* < 0.001 show statistical significance compared to the group of cadmium-intoxicated rats. Values are stated as the mean ± standard error of the mean. LIN, linagliptin; Cd, cadmium chloride.

**Figure 6 pharmaceuticals-15-00852-f006:**
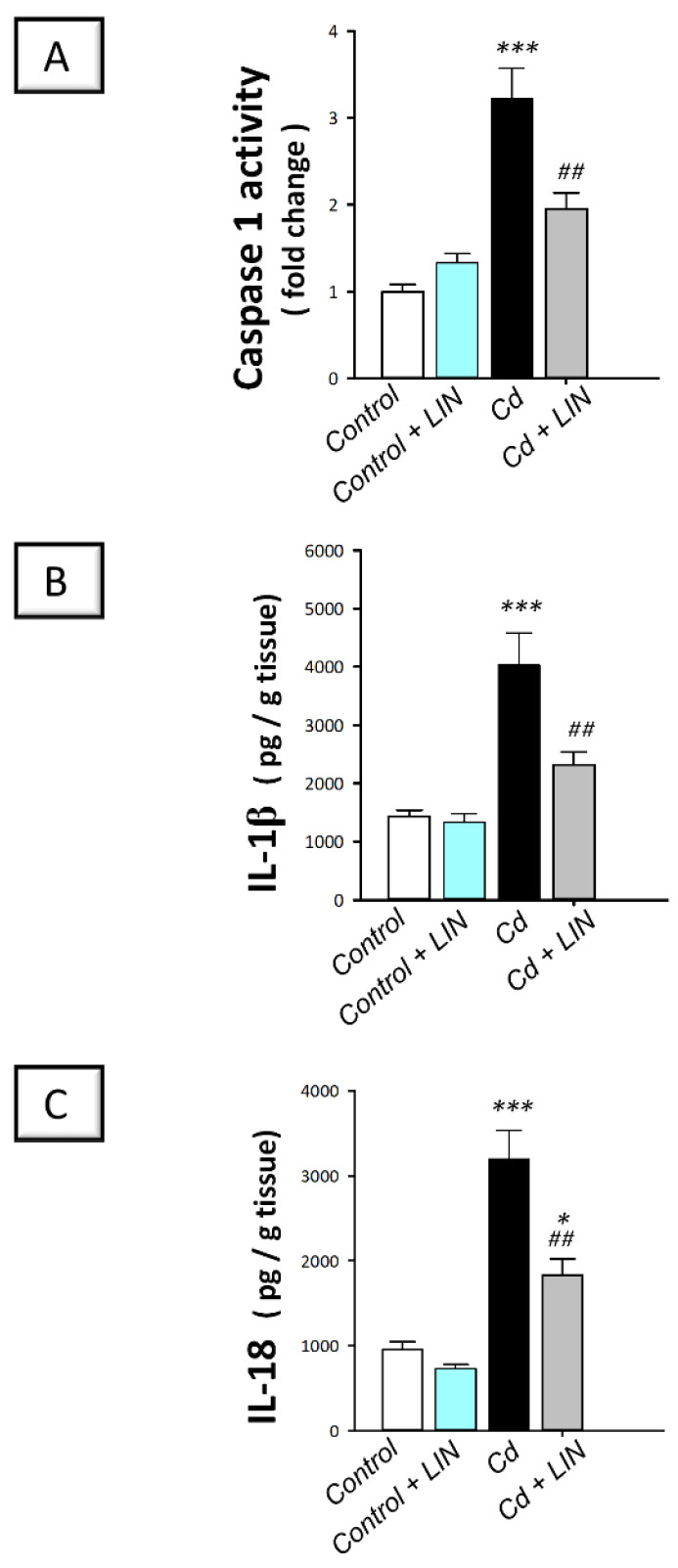
Effect of cadmium and/or linagliptin administration on the testicular caspase-1/IL-1β/IL-18 pathway in rats. The co-treatment with linagliptin inhibited the testicular caspase-1/IL-1β/IL-18 pathway that was stimulated by cadmium metal. This was demonstrated by diminished caspase 1 activity (**A**) together with lowered levels of interleukin 1 beta (IL-1β; (**B**)), and interleukin 18 (IL-18; (**C**)). ** p* < 0.05, and **** p* < 0.001 show statistical significance compared to the group of control rats; *^##^ p* < 0.01 show statistical significance compared to the group of cadmium-intoxicated rats. Values are stated as the mean ± standard error of the mean for 6 animals per group. LIN, linagliptin; Cd, cadmium chloride.

**Figure 7 pharmaceuticals-15-00852-f007:**
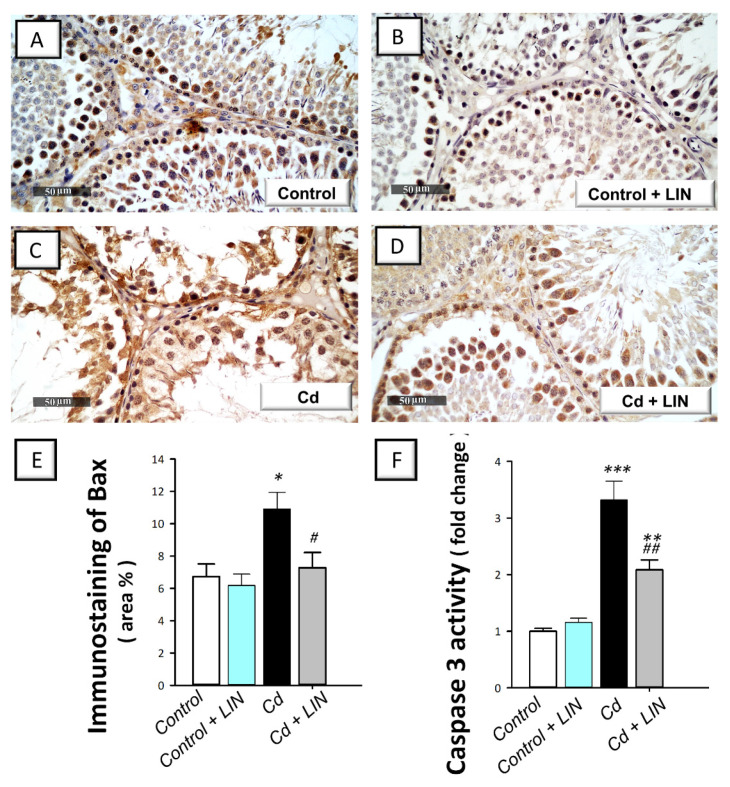
Effect of cadmium and/or linagliptin administration on testicular apoptosis in rats. The co-treatment with linagliptin curbed testicular pro-apoptotic events that were turned on by cadmium metal. This was demonstrated by downregulated protein expression of Bcl-2 associated x protein (Bax) as detected by immunohistochemistry that shows up as a brown color at 400× magnification. (**A**,**B**) The immunostaining of testicular Bax protein demonstrates minimal expression in the control and control + LIN groups. (**C**) The immunostaining of the pro-apoptotic Bax was upregulated in the cadmium-intoxicated group. (**D**) The immunostaining of Bax was downregulated in the Cd + LIN group. (**E**) Quantitative analysis of Bax immunostaining in the testicular tissue presented as the area % of the immunoreactivity/full microscopic field area. Six non-overlapping fields were inspected for each specimen. In addition, co-treatment with linagliptin diminished the activity of testicular executioner caspase 3 (**F**). ** p* < 0.05, *** p* < 0.01, and **** p* < 0.001 show statistical significance compared to the group of control rats; *^#^ p* < 0.05, and *^##^ p* < 0.01 show statistical significance compared to the group of cadmium-intoxicated rats. Values are stated as the mean ± standard error of the mean for n = 6 per group. LIN, linagliptin; Cd, cadmium chloride.

**Figure 8 pharmaceuticals-15-00852-f008:**
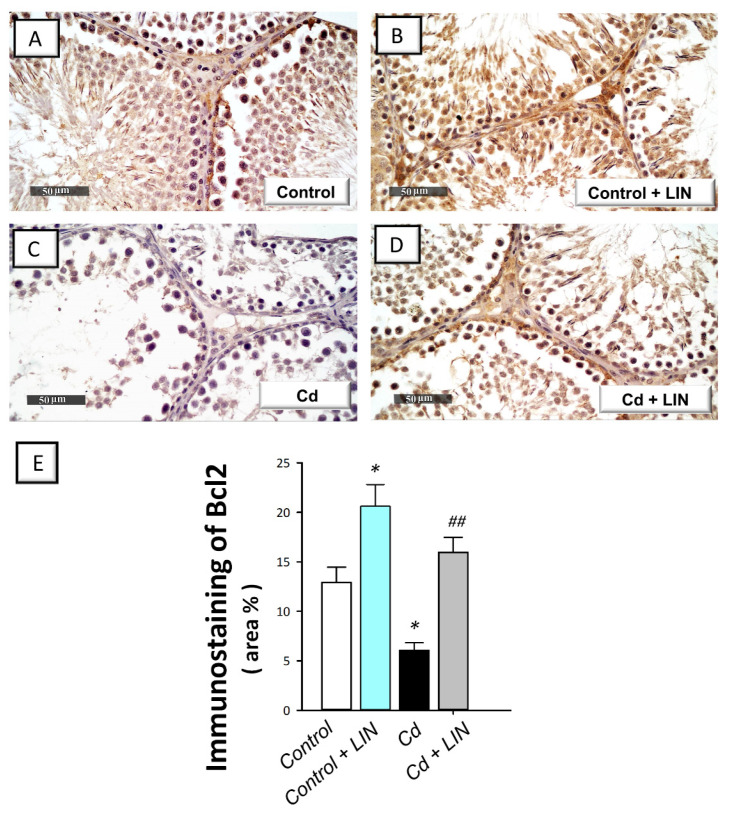
Effect of cadmium and/or linagliptin administration on testicular Bcl-2 protein expression in rats. The co-treatment with linagliptin intensified the testicular protein expression of B-cell lymphoma-2 (Bcl-2) that was downregulated by cadmium metal as detected by immunohistochemistry that shows up as brown immunostaining at 400× magnification. (**A**,**B**) The immunostaining of testicular Bcl-2 protein demonstrates intense expression in the control and control + LIN groups. (**C**) The immunostaining of the pro-apoptotic Bcl-2 was downregulated in the cadmium-intoxicated group. (**D**) The immunostaining of Bcl-2 was upregulated in the Cd + LIN group. (**E**) Quantitative analysis of Bcl-2 immunostaining in the testicular tissue presented as the area % of the immunoreactivity/full microscopic field area. Six non-overlapping fields were inspected for each specimen. ** p* < 0.05 show statistical significance compared to the group of control rats; *^##^ p* < 0.01 show statistical significance compared to the group of cadmium-intoxicated rats. Values are stated as the mean ± standard error of the mean. LIN, linagliptin; Cd, cadmium chloride.

**Figure 9 pharmaceuticals-15-00852-f009:**
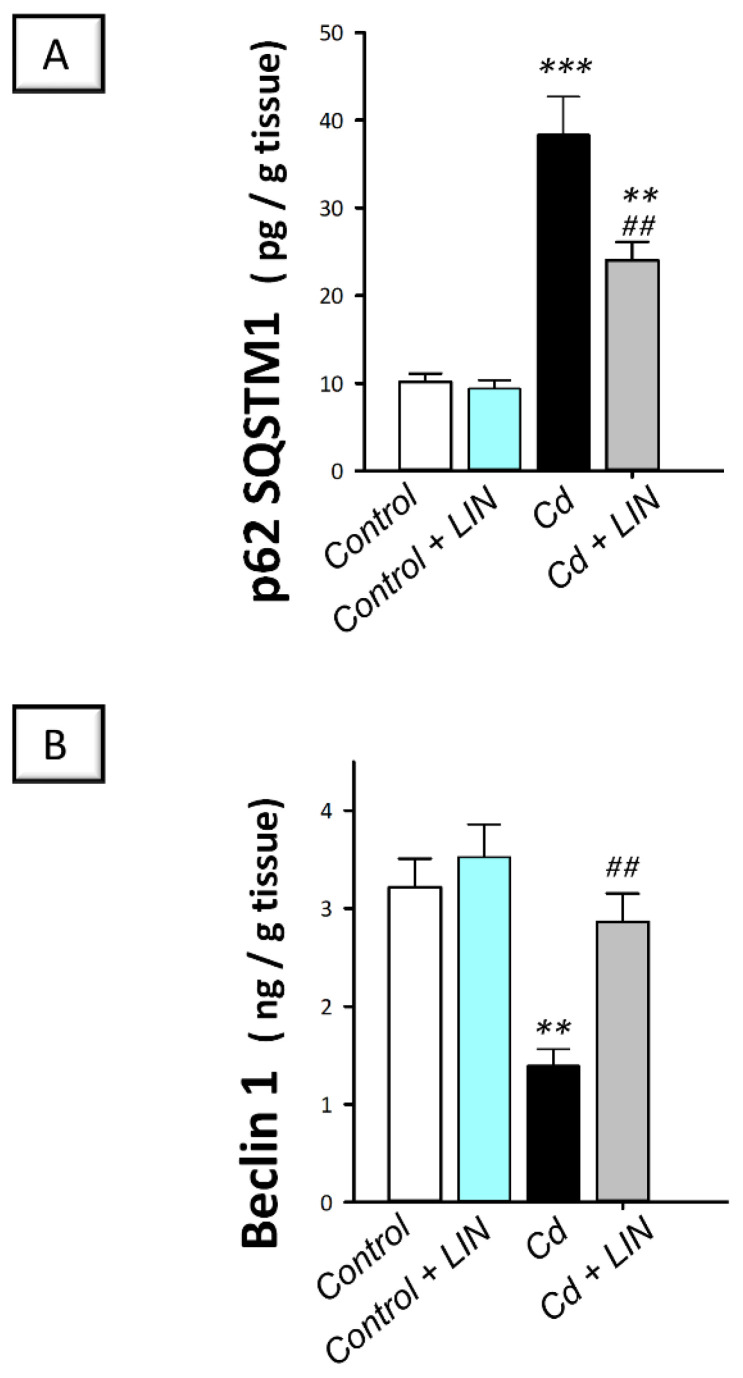
Effect of cadmium and/or linagliptin administration on testicular autophagy in rats. The co-treatment with linagliptin enhanced the testicular autophagy response that was compromised by cadmium metal. This was demonstrated by diminished protein expression of the negative autophagy marker protein 62 sequestome 1 (p62 SQSTM1; (**A**)) alongside increased protein expression of the positive autophagy marker Beclin 1 (**B**). *** p* < 0.01, and **** p* < 0.001 show statistical significance compared to the group of control rats; *^##^ p* < 0.01 show statistical significance compared to the group of cadmium-intoxicated rats. Values are stated as the mean ± standard error of the mean for 6 animals per group. LIN, linagliptin; Cd, cadmium chloride.

**Figure 10 pharmaceuticals-15-00852-f010:**
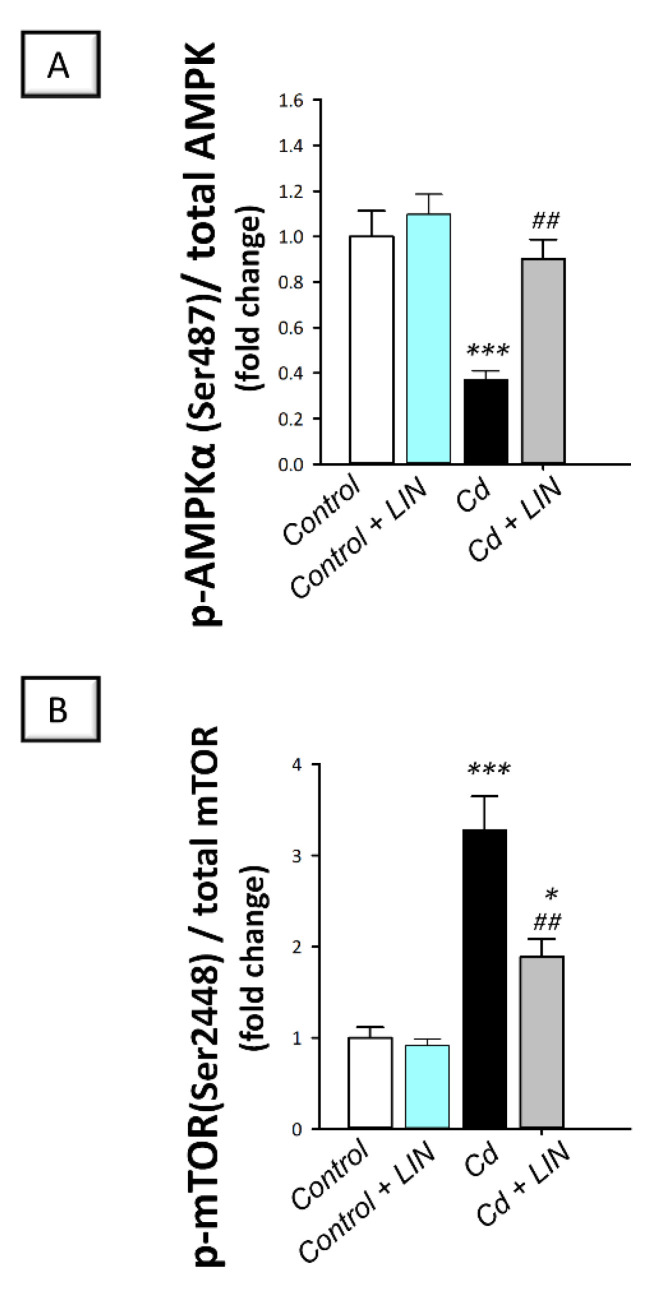
Effect of cadmium and/or linagliptin administration on testicular AMPK/mTOR in rats. The co-treatment with linagliptin stimulated the testicular pro-autophagic AMPK/mTOR pathway that was suppressed by cadmium metal. This was demonstrated by increased phosphorylation of the AMP-activated protein kinase (AMPK; Ser487) to the total AMPK protein (**A**) together with the diminished phosphorylation of the mammalian target of rapamycin mTOR (Ser2448) to the total mTOR protein (**B**). ** p* < 0.05, and **** p* < 0.001 show statistical significance compared to the group of control rats; *^##^ p* < 0.01 show statistical significance compared to the group of cadmium-intoxicated rats. Values are stated as the mean ± standard error of the mean for 6 animals per group. LIN, linagliptin; Cd, cadmium chloride.

**Figure 11 pharmaceuticals-15-00852-f011:**
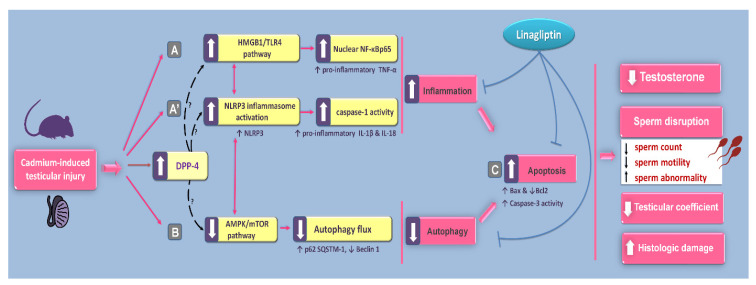
An outline of the mechanisms that intercede linagliptin’s amelioration of cadmium-evoked testicular impairment. According to the current findings, the proposed mechanisms are: (**A**,**A′**) Curbing the inflammatory responses, including HMGB1/TLR4/NF-κB and NLRP3/caspase-1/IL-1β pathways in the testes of cadmium-intoxicated rats. (**B**) Stimulation of the autophagy response with activation of AMPK/mTOR pathway in the testes of cadmium-intoxicated rats. (**C**) Interference with the pro-apoptotic responses, including Bax downregulation, Bcl2 upregulation, and caspase 3 inhibition in the testes of cadmium-intoxicated rats. Of note, the beneficial outcomes of linagliptin are mainly due to DPP-4 inhibition, which has been linked to marked anti-inflammatory/immune-modulatory actions [21,22]. DPP-4 inhibition triggers several downstream targets, including the upregulated protein expression of SDF-1α, which prompts intensified testicular repair and regeneration [21,33]. Moreover, the beheld activation of autophagy may suppress the pro-inflammatory responses/NLRP3 inflammasome activation [12,20] and limit the proapoptotic responses [12,17,19] in the testes of cadmium-intoxicated rats. Activation is represented by solid arrows, whereas inhibition is depicted by blunt arrows. The question marks denote unknown mechanisms.

**Figure 12 pharmaceuticals-15-00852-f012:**
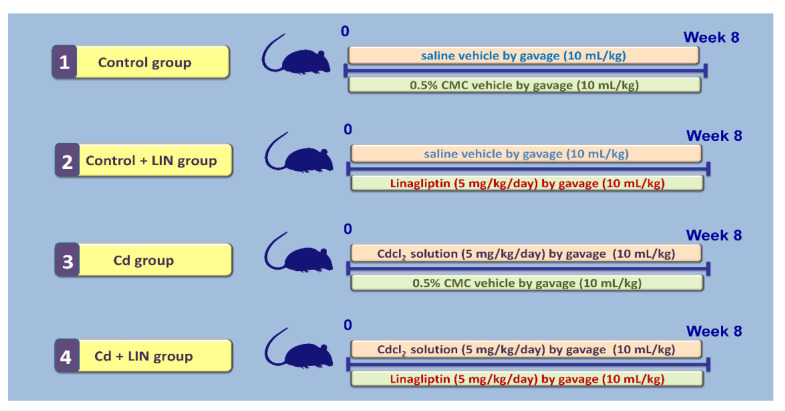
An outline of the current experimental protocol. LIN, linagliptin; Cd, cadmium chloride; CMC, 0.5% carboxymethyl cellulose.

## Data Availability

Data are contained within the article.

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
