# Peer review of "Repositioning Linagliptin for the Mitigation of Cadmium-Induced Testicular Dysfunction in Rats: Targeting HMGB1/TLR4/NLRP3 Axis and Autophagy"

_pharmaceuticals, 2022, doi:10.3390/ph15070852_

Round 1

Reviewer 1 Report

The authors aimed to verify the effects of cadmium in the testicular function and apoptosis and if linagliptin is able to restore it. 

The authors contructed a good hypothesis in the introduction and presented the results in a good and simple manner, facilitating the understading to the readers. 

Still, there are some questions that should be addressed regarding the methods section:

- Section 4.3: Study design is very confusing when describing the groups construction. You mention cadmium gavage and linagliptin in all of them. They should be better explained. 

- In addition the whole section, It was a little confusing if the markers was measured in blood, semen, testis paraffined or testis homogenates. In envery experiment section it should be mentioned which material was used. 

Author Response

Dear Editor and referees,

We are enclosing a revised manuscript for your consideration for publication in Pharmaceuticals. We would like to thank the referees, the editor, and the editor-in-chief for closely reading our manuscript and for the very helpful comments which significantly guided us to improve our work. We have made revisions to the manuscript per the critique and addressed all the reviewers' comments. All the modifications to the manuscript are highlighted in light blue in the revised version. The details for the modifications of the manuscript in response to the reviewers are listed as follows. 

Thank you again for your time. We look forward to hearing from you.

Kind Regards,

Hany Arab, Ph.D., the corresponding author

Response to reviewer #1:

We thank the reviewer for the insightful comments regarding our study that guided us in improving the manuscript.

Comment 1: The authors aimed to verify the effects of cadmium in the testicular function and apoptosis and if linagliptin is able to restore it. The authors constructed a good hypothesis in the introduction and presented the results in a good and simple manner, facilitating the understanding to the readers”. 

Response: We thank the referee for taking the time to closely review the manuscript and for commending our manuscript.

Comment 2: “Still, there are some questions that should be addressed regarding the methods section. Section 4.3: Study design is very confusing when describing the groups construction. You mention cadmium gavage and linagliptin in all of them. They should be better explained”. 

Response: We thank the expert reviewer for the perceptive comment. This point has been clarified on the Material and Methods section as follows:

  1. A) In section 4.3 (Experimental design and protocol), we have revised the text and clarified the writing to clearly describe each experimental group. This part has been modified in page 17, lines: 51729-540, as follows:

As depicted in Figure 12, the first group (Control group; 6 animals) received a daily gavage of the vehicle of cadmium chloride (normal saline; 10 mL/kg) plus a daily gavage of linagliptin vehicle (0.5% carboxymethyl cellulose; CMC; 10 mL/kg, 1h after cadmium chloride vehicle) for 8 weeks. The second group (Control + LIN group; 6 animals) received a daily gavage of normal saline vehicle (10 mL/kg) plus a daily gavage of linagliptin (5 mg/kg; 1h after cadmium chloride vehicle) suspended in CMC vehicle (10 mL/kg) for 8 weeks. The third group (Cd group; 6 animals) received a daily gavage of cadmium chloride (5 mg/kg in normal saline; 10 mL/kg) plus a daily gavage of CMC vehicle (1h after cadmium chloride solution) for 8 weeks. The fourth group (Cd + LIN group; 6 animals) received a daily gavage of cadmium chloride (5 mg/kg in normal saline; 10 mL/kg) plus a daily gavage of linagliptin (5 mg/kg; 1h after cadmium chloride solution) suspended in CMC vehicle (10 mL/kg) for 8 weeks”.

  1. B) We have added figure 12 to graphically illustrate the experimental design of the present work in a simple way as follows:

Figure 12. An outline of the current experimental protocol. LIN, linagliptin; Cd, cadmium chloride; CMC, 0.5% carboxymethyl cellulose.  

Comment 3: “In addition the whole section, It was a little confusing if the marker was measured in blood, semen, testis paraffined, or testis homogenates. In every experiment section, it should be mentioned which material was used”.

Response: We thank the reviewer for the insightful comment. Per the reviewer’s suggestion, this point has been addressed as follows:

  1. A) In the material and methods section, we have clearly described under each experimental section whether each marker was measured in blood, semen, paraffined testis, or testis homogenate. Please, refer to the revised version of the manuscript.

  1. B) In the results section and in all figure legends, we have clearly described where the markers were measured. Please, refer to the revised version of the manuscript.

Reviewer 2 Report

 1)     Ok, you observed with linagliptin a reduction in the levels of DPP-4, but could you tell me how is the enzymatic activity of DPP-4? Is the DPP-4 activity increased in the Cd+LIN group by having more SDF-1alpha substrate to hydrolyze?

2)     There are two types of DPP-4, the soluble and the membrane-bound form. Does linagliptin inhibit both equally or only one subtype?

3)     Thus, linagliptin downregulates the entire HMGB1/TLR4/NLRP3 axis and inhibits its subsequent proinflammatory and apoptotic processes (as well as being proautophagic). Therefore, is it known exactly if it is a direct inhibition on each component of the axis? Or can there also be an intermediate step that is unknown?

4)     Regarding the above, what relationship does the DPP-4 have with this axis? Does it have any interaction with the catalytic domain of DPP-4 or possibly with the ADA binding domain of lymphocytes and monocytes?

Author Response

Dear Editor and referees,

We are enclosing a revised manuscript for your consideration for publication in Pharmaceuticals. We would like to thank the referees, the editor, and the editor-in-chief for closely reading our manuscript and for the very helpful comments which significantly guided us to improve our work. We have made revisions to the manuscript per the critique and addressed all the reviewers' comments. All the modifications to the manuscript are highlighted in light blue in the revised version. The details for the modifications of the manuscript in response to the reviewers are listed as follows. 

Thank you again for your time. We look forward to hearing from you.

Kind Regards,

Hany Arab, Ph.D., the corresponding author

Response to reviewer #2:

We thank the reviewer for the insightful comments regarding our study that guided us for improving the manuscript.

Comment 1: Ok, you observed with linagliptin a reduction in the levels of DPP-4, but could you tell me how is the enzymatic activity of DPP-4? Is the DPP-4 activity increased in the Cd+LIN group by having more SDF-1alpha substrate to hydrolyze?”. 

Response: We thank the reviewer for the insightful comment. This point has been addressed as follows:

  1. A) In response to the most potent DPP-4 inhibitor linagliptin, the current work indirectly measured the activity of DPP-4 through investigating the levels of stromal cell-derived factor-1 alpha (SDF-1α), a substrate for DPP-4, in the testicular tissue. We observed that DPP-4 inhibition by linagliptin increased the protein expression of SDF-1α, confirming the effective inhibition of DPP-4 enzyme (Figure 3B).

  1. B) Meanwhile, the present work quantified the protein expression of DPP-4 in testicular tissues of rats that was effectively downregulated by linagliptin. In fact, previous studies demonstrated that DPP-4 inhibition by gliptins lowers the protein expression alongside the enzymatic activity of DPP-4 in rodents in vivo (Kanasaki, 2018 and Benetti et al., 2021). In this context, a correlation between the enzymatic activity of DPP-4 and its protein expression was characterized. Notably, effective inhibition of DPP-4 was also accompanied by downregulated protein expression of DPP-4.

-------------------------------------------------------

References

Acaris Benetti, Flavia Letícia Martins, Letícia Barros Sene, Maria Heloisa M. Shimizu,2 Antonio C. Seguro, Weverton M. Luchi, Adriana C. C. Girardi. Urinary DPP4 correlates with renal dysfunction, and DPP4 inhibition protects against the reduction in megalin and podocin expression in experimental CKD. Am J Physiol Renal Physiol 2021, 320(3):F285-F296. doi: 10.1152/ajprenal.00288.2020.

Kanasaki, K. The role of renal dipeptidyl peptidase-4 in kidney disease: renal effects of dipeptidyl peptidase-4 inhibitors with a focus on linagliptin. Clin Sci (Lond) 2018, 132, 489-507, doi:10.1042/CS20180031.

-------------------------------------------------------

  1. C) In fact, the current work is a proof-of-concept study that pointed to the efficacy of linagliptin in mitigating cadmium-induced testicular dysfunction. In the present work, we have focused on the efficacy of linagliptin in mitigating cadmium-induced testicular injury at the microscopic levels where it diminished the immune-cell infiltration and histopathologic aberrations. The favorable effects of linagliptin were mediated by downregulating the protein expression of HMGB1, RAGE, and nuclear NF-κBp65 pro-inflammatory signals alongside its downstream signals, including TNF-α. Moreover, we have detected the alterations of Beclin 1 and p62 SQSTM1 autophagy markers and explored the AMPK/mTOR pathway by detecting the expression of p-AMPK/AMPK and the autophagy suppressor p-mTOR/mTOR signal. In addition to the autophagy signals, we have investigated the apoptotic changes, including the protein expression of Bax, Bcl-2, and the Bax/Bcl-2 ratio, and the activity of caspase-3 activity.

  1. D) Future studies are needed to address the detailed molecular mechanisms of linagliptin against cadmium-induced testicular injury, including its impact on DPP-4 activity. This point has been clarified in page 20, lines: 676-680) as follows:

“Hence, additional studies are warranted to delineate the detailed molecular mechanisms of linagliptin. Meanwhile, further investigations are needed to characterize whether the anti-inflammatory/anti-apoptotic actions of linagliptin are driven by direct inhibition on each component of the HMGB1/TLR4/NLRP3 axis or other intermediate steps are also involved”.

Comment 2: “There are two types of DPP-4, the soluble and the membrane-bound form. Does linagliptin inhibit both equally or only one subtype?”

Response: We thank the reviewer for the perceptive comment. DPP-4 is a ubiquitous protein with exopeptidase activity that exists in membrane-bound form within tissues in addition to the soluble form found in plasma and other body fluids. The literature revealed that linagliptin was effective in suppressing the activity of the two forms of DPP-4 as demonstrated in an ex vivo study in Zucker diabetic fatty (ZDF) rats. This point has been addressed in the introduction section (page: 3; lines: 124-126) as follows:

“DPP-4 is an exopeptidase enzyme that occurs in two forms; membrane-anchored within tissues and soluble form in body fluids including plasma [25]. Interestingly, linagliptin has been demonstrated to inhibit both forms of DPP-4 [26]”.

Comment 3: “Thus, linagliptin downregulates the entire HMGB1/TLR4/NLRP3 axis and inhibits its subsequent proinflammatory and apoptotic processes (as well as being proautophagic). Therefore, is it known exactly if it is a direct inhibition on each component of the axis? Or can there also be an intermediate step that is unknown?”

Response: We thank the reviewer for the insightful comment. This point has been addressed as follows:

  1. A) We have described some underlying mechanisms which were described in previous literature. This point has been addressed in page: 20, lines: 665-673, as follows:

“These beneficial effects are mainly driven by DPP-4 inhibition which has been linked to marked anti-inflammatory and immune modulatory effects seen in several experimental testicular injury models [21,22]. In response to DPP-4 inhibition, previous reports revealed that suppression of the pro-inflammatory events by linagliptin is ensued likely through disrupting the interaction between membrane-anchored DPP-4 and integrin β1 alongside the modulation of mi-RNA-29 [26]. In the same regard, DPP-4 inhibition in T-lymphocytes has been demonstrated to disrupt the protein-protein interaction between cysteine-rich domain of DPP-4 and adenosine deaminase (ADA), culminating in dampened pro-inflammatory cytokine production [63]”.

  1. B) However, evidence is lacking whether a direct interaction between linagliptin and pro-inflammatory signals including HMGB1, TLR4, and NLRP3. Hence, the exact molecular mechanisms warrant further investigation in order to define whether the downregulation of inflammatory events is a direct inhibition on each component of the HMGB1/TLR4/NLRP3 axis or they are due to an intermediate step. This point has been addressed in page: 20, lines: 676-680, as follows:

“Hence, additional studies are warranted to delineate the detailed molecular mechanisms of linagliptin. Meanwhile, further investigations are needed to characterize whether the anti-inflammatory/anti-apoptotic actions of linagliptin are driven by direct inhibition on each component of the HMGB1/TLR4/NLRP3 axis or other intermediate steps are also involved”.

Comment 4: “Regarding the above, what relationship does the DPP-4 have with this axis? Does it have any interaction with the catalytic domain of DPP-4 or possibly with the ADA binding domain of lymphocytes and monocytes?”

Response: We thank the reviewer for the perceptive comment. Previous reports have revealed that binding of linagliptin to membrane-bound DPP-4 disrupts the interaction between the cysteine-rich domain of DPP-4 and adenosine deaminase (ADA). This event is associated with suppressed generation of pro-inflammatory signals including cytokines. This point has been addressed in page: 20, lines: 669-672, as follows:

“In the same regard, DPP-4 inhibition in T-lymphocytes has been demonstrated to disrupt the protein-protein interaction between cysteine-rich domain of DPP-4 and adenosine deaminase (ADA), culminating in dampened pro-inflammatory cytokine production [63]”.
